# Comparison of Efficacy and Safety of Platinum-Based Chemotherapy as First-Line Therapy between B3 Thymoma and Thymic Carcinoma

Yue Hao [1,2,†] , Jinfei Si [1,2,†] , Jianan Jin [2,†] , Jingwen Wei [2], Jing Xiang [1,2], Chunwei Xu [3]
and Zhengbo Song [2,*]

1    The Second Clinical Medical College of Zhejiang Chinese Medical University, Hangzhou 310000, China
2    Department of Clinical Trial, Cancer Hospital of the University of Chinese Academy of Sciences, Zhejiang Cancer Hospital, Hangzhou 310022, China
3    Department of Respiratory Medicine, Jinling Hospital, School of Medicine, Nanjing University, Nanjing 210093, China
\*    Correspondence: songzb@zjcc.org.cn
†    These authors contributed equally to this work.

**Abstract:** Background: B3 type thymoma is defined as a well-differentiated thymic carcinoma and is similar to a thymic carcinoma. However, the differences between them are not well defined. In addition, the data to compare the efficacy and safety of platinum-based chemotherapy as first-line therapy between B3 thymoma and thymic carcinoma are lacking. Methods: The efficacy and safety of platinum-based chemotherapy as first-line therapy was retrospectively compared between a group of 36 patients with type B3 thymoma and a group of 127 patients with thymic carcinoma treated between January 2009 and March 2022 at the Zhejiang Cancer Hospital. Objective response rate (ORR), progression-free survival (PFS), overall survival (OS), and treatment-related adverse events were analyzed. Results: The ORRs for B3 thymoma and thymic carcinoma were 36.1% and 28.3%, respectively ($p = 0.370$). Among all patients, the difference in PFS between B3 thymoma and thymic carcinoma was not significant (11.3 vs. 10.1 months, $p = 0.118$). The squamous carcinoma subgroup did not exhibit any differences in PFS compared to B3 thymoma (11.7 vs. 11.3 months, $p = 0.161$). The result for the non-squamous carcinoma subgroup was similar (6.5 vs. 11.3 months, $p = 0.128$). Furthermore, the OS values for B3 thymoma and thymic carcinoma were not significantly different (58.3 vs. 35.1 months, $p = 0.067$). However, there were differences in OS between B3 thymoma and non-squamous carcinoma (58.3 vs. 30.6 months, $p = 0.031$). Conclusions: B3 thymoma and especially squamous carcinoma patients may be treated using a similar therapy scheme as that utilized for thymic carcinoma.

**Keywords:** B3 thymoma; thymic carcinoma; platinum-based chemotherapy; efficacy; safety

## 1. Introduction

Thymic epithelial tumor is a rare cancer type occurring in the mediastinum [1]. Its incidence rate ranges between 0.9 and 2.3 cases per million people [2]. According to the histologic classification, thymic epithelial tumors are divided into different subtypes, including A, AB, B1, B2, B3, and C. Due to the high rate of relapse and metastasis, the C type thymoma is defined as a thymic carcinoma with a worse prognosis [3,4]. For advanced thymoma and thymic carcinoma, systematic chemotherapy represents a multimodality approach that is recommended as the standard therapy [5].

The degree of malignancy increases with the increase in subtype grade [6]. As a borderline tumor between thymoma and thymic carcinoma, B3 type thymoma is defined as a well-differentiated thymic carcinoma or squamoid thymoma. There is no clear distinction in clinical guidelines for therapy efficacy between thymic carcinoma and B3 thymoma.

Furthermore, little is known about the differences in survival between the two. Whether B3 squamoid thymoma patients can receive clinical management and treatment similar to that used for thymic squamous carcinoma remains uncertain. In addition, data to compare the efficacy of chemotherapy for these two types are lacking.

The current study aimed to analyze the relationship between B3 thymoma and thymic carcinoma based on the survival data, and to compare the efficacy and safety of platinum-based chemotherapy as first-line therapy.

## 2. Materials and Methods

### 2.1. Study Design

Patients with clinical stage III (not suitable for curative surgery or radiotherapy) and IV thymic carcinoma and B3 thymoma who had undergone treatment at the Zhejiang Cancer Hospital between January 2009 and March 2022 were retrospectively identified. The World Health Organization classification was used to determine the histological types. Other inclusion criteria were as follows: measurable lesions defined by Response Evaluation Criteria in Solid Tumors 1.1, ECOG (Eastern Cooperative Oncology Group) performance status 0 or 1, adequate bone marrow reserve, and normal hepatic and renal functions. All patients received platinum-based chemotherapy as first-line therapy. During the period of first-line therapy, other drugs were not administered. Cases with prior malignancies were excluded from the study. The Institutional Ethics Committee at the Zhejiang Cancer Hospital provided the study protocol for research. Individual consent for analysis was waived.

### 2.2. Treatment and Response Assessments

The efficacy and safety of platinum-based chemotherapy as first-line therapy were included in the analysis. All types of platinum-based regimens were incorporated. All patients received four to six cycles of chemotherapy. The regimens were continued if the patients could benefit from the therapy. Outcomes were assessed by computed tomography scans after the second and fourth cycles and at the end of treatment (every six weeks). The safety was assessed according to the National Cancer Institute Common Terminology Criteria (NCI-CTC) for Adverse Events (AEs), version 4.03. If treatment-related AEs or severe disease progression occurred, the computed tomography scan could also be taken.

The objective response rate (ORR) was defined as the sum of the complete response and partial response. Progression-free survival (PFS) encompassed the time from the first chemotherapy treatment administration to the documented disease progression or death for any reason. Overall survival (OS) was determined from the date of first platinum-based chemotherapy administration to death or last follow-up evaluation.

### 2.3. Statistical Analysis

Basic patient characteristics and distinctions among different subgroups were analyzed using a chi-squared test. Statistical analysis was performed using SPSS version 25 (SPSS Inc., Chicago, IL, USA), assuming that $p < 0.05$ is statistically significant. The GraphPad Prism (version 9) was also used to analyze the survival outcomes. The Kaplan–Meier method and GraphPad Prism were used to plot the survival curves. The difference in survival between different subgroups was evaluated according to the outcomes of log-rank analysis. The last follow-up date was 28 March 2022.

## 3. Results

### 3.1. Patients

A total of 163 patients, including 36 B3 type thymoma and 127 thymic carcinoma cases were identified and incorporated into analysis. The thymic carcinoma group was divided into two subgroups, where squamous carcinoma patients accounted for 64.6% of all cases. The patient characteristics are listed in Table 1. The age range of all patients was 20–73 years. The study included 106 men and 57 women. Furthermore, 159 patients had

stage IV disease. All patients had an ECOG performance status of 0 or 1. There were no notable differences in sex, age, stage, smoking history, and radiotherapy history between the two groups. All patients received platinum-based chemotherapy as first-line therapy.

**Table 1.** Patients characteristics.

| Characteristics | B3 Thymoma (n = 36) | | Thymic Carcinoma (n = 127) | | p Value |
|---|---|---|---|---|---|
| | No. | % | No. | % | |
| Sex | | | | | 0.155 |
| Male | 27 | 75.0 | 79 | 62.2 | |
| Female | 9 | 25.0 | 48 | 37.8 | |
| Age | | | | | 0.095 |
| Median | 51.5 | | 57 | | |
| Range | 22–69 | | 20–73 | | |
| ≤65 | 34 | 94.4 | 106 | 83.5 | |
| >65 | 2 | 5.6 | 21 | 16.5 | |
| Stage | | | | | 1.000 |
| III | 1 | 2.8 | 3 | 2.4 | |
| IV | 35 | 97.2 | 124 | 97.6 | |
| Smoking history | | | | | 0.071 |
| Former | 22 | 61.1 | 56 | 44.1 | |
| Never | 14 | 38.9 | 71 | 55.9 | |
| ECOG PS | | | | | 0.670 |
| 0 | 21 | 58.3 | 69 | 54.3 | |
| 1 | 15 | 41.7 | 58 | 45.7 | |
| Surgery | | | | | 0.011 |
| Yes | 18 | 50.0 | 35 | 27.6 | |
| No | 18 | 50.0 | 92 | 72.4 | |
| Radiotherapy | | | | | 0.340 |
| Yes | 21 | 58.3 | 85 | 66.9 | |
| No | 15 | 41.7 | 42 | 33.1 | |

Abbreviations: ECOG PS, Eastern Cooperative Oncology Group Performance status.

*3.2. Treatment Response and Survival Analyses*

Thirteen patients with B3 thymoma treated with chemotherapy achieved a partial response (PR) and 20 exhibited stable disease (SD) compared to 36 patients with PR and 78 with SD in the thymic carcinoma group. There were no complete responders. As the majority index of treatment response, ORR was 36.1% for B3 thymoma patients compared to 28.3% for thymic carcinoma patients ($p = 0.370$; Table 2).

**Table 2.** Response Rates for the Intent-to-Treat B3 thymoma and Thymic carcinoma Population.

| Response Rates | B3 Thymoma (n = 36) | | Thymic Carcinoma (n = 127) | | p Value |
|---|---|---|---|---|---|
| | No. | % | No. | % | |
| Overall response | 13 | 36.1 | 36 | 28.3 | 0.370 |
| complete response | 0 | 0 | 0 | 0 | |
| partial response | 13 | 36.1 | 36 | 28.3 | |
| stable disease | 20 | 55.6 | 78 | 61.5 | |
| progressive disease | 3 | 8.3 | 13 | 10.2 | |

The PFS for B3 thymoma was 11.3 months, which was not significantly different compared to 10.1 months in thymic carcinoma patients ($p = 0.118$; Figure 1a). The OS results were similar (58.3 vs. 35.1 months, $p = 0.067$; Figure 1b).

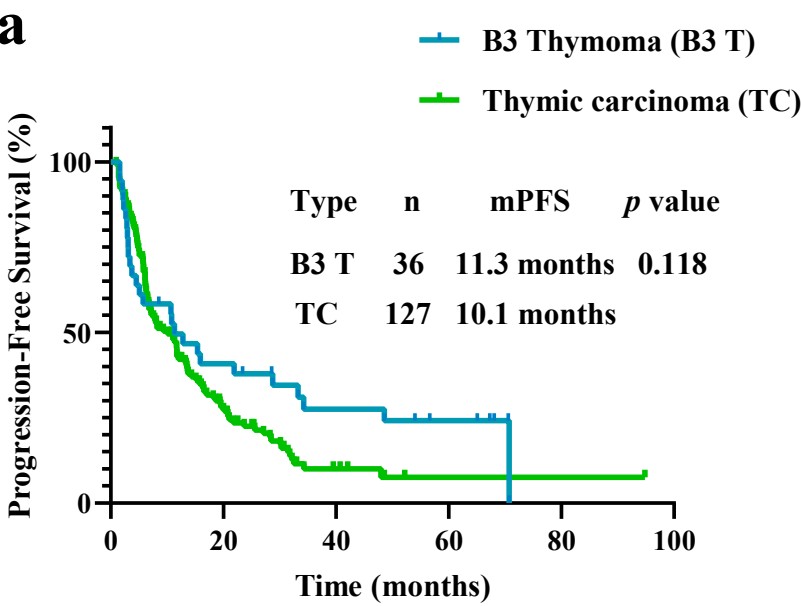

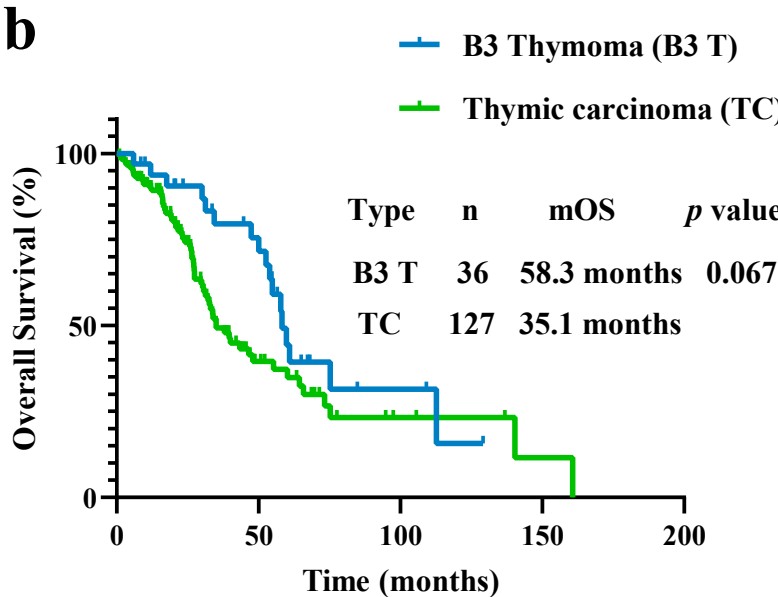

**Figure 1.** (**a**) Progression-free survival (PFS) and (**b**) overall survival (OS) in patients with B3 thymoma and thymic carcinoma. PFS and OS did not differ between patients with B3 thymoma and thymic carcinoma (PFS, 11.3 vs. 10.1 months, *p* = 0.118; OS, 58.3 vs. 35.1 months, *p* = 0.067).

In the squamous carcinoma subgroup, the PFS was not different compared to B3 thymoma patients (11.7 vs. 11.3 months, *p* = 0.161; Figure 2a). The OS for squamous carcinoma patients was 40.0 months, which was not different compared to the B3 thymoma patients' value of 58.3 months (*p* = 0.114; Figure 2b).

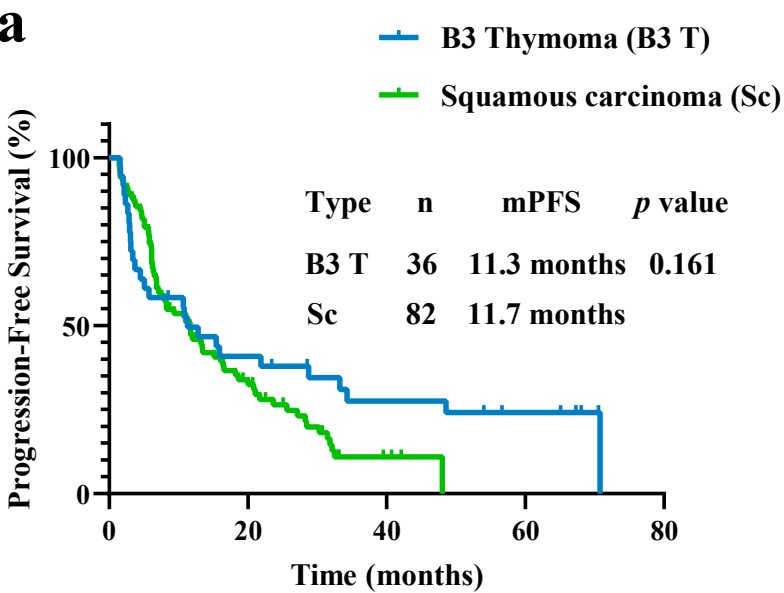

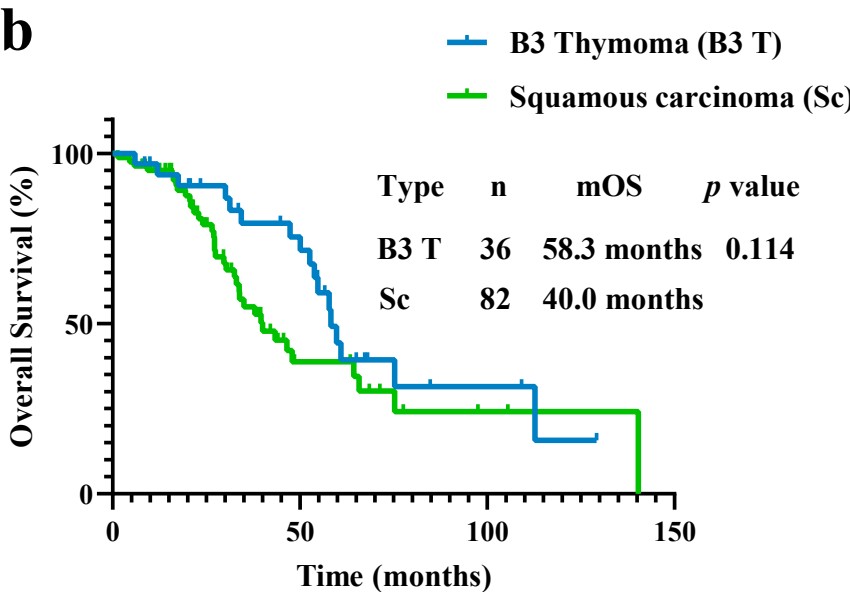

**Figure 2.** (**a**) PFS and (**b**) OS in patients with B3 thymoma and squamous carcinoma. PFS and OS did not differ between patients with B3 thymoma and squamous carcinoma (PFS, 11.3 vs. 11.7 months, $p = 0.161$; OS, 58.3 vs. 40.0 months, $p = 0.114$).

The PFS was 6.5 months in the non-squamous carcinoma subgroup, which was lower than 11.3 months in the B3 thymoma group although the difference was not statistically significant ($p = 0.128$; Figure 3a). However, a difference in OS between non-squamous carcinoma and B3 thymoma was observed (30.6 vs. 58.3 months, $p = 0.031$; Figure 3b).

Different therapy regimens did not exhibit any differences in survival. Patients who received paclitaxel regimen had a similar PFS compared to patients receiving non-paclitaxel regimen (11.4 vs. 6.8 months, $p = 0.689$). The OS results were similar (43.4 vs. 47.4 months, $p = 0.758$).

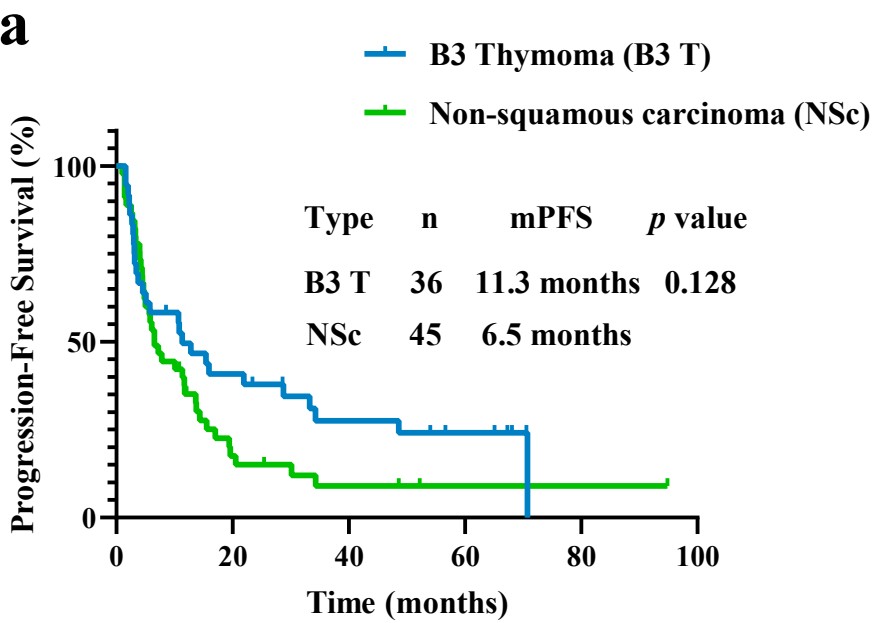

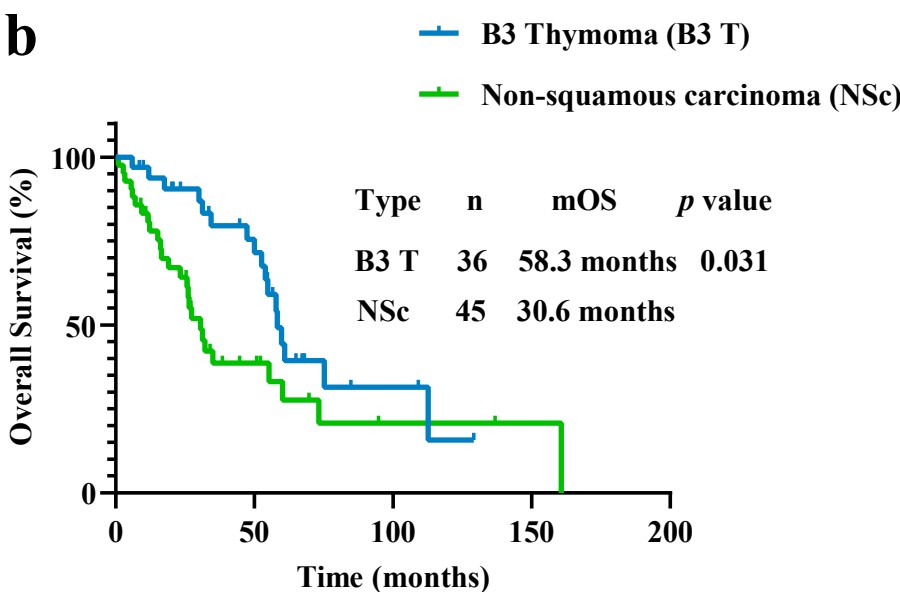

**Figure 3.** (**a**) PFS and (**b**) OS in patients with B3 thymoma and non-squamous carcinoma. PFS did not differ between patients with B3 thymoma and non-squamous carcinoma (PFS, 11.3 vs. 6.5 months, $p = 0.128$). OS was different between patients with B3 thymoma and non-squamous carcinoma (OS, 58.3 vs. 30.6 months, $p = 0.031$).

### 3.3. Toxicity

Neutropenia was observed in both groups, which accounted for 30.6% and 25.2% of patients in each group, respectively ($p = 0.520$). Among all patients, three patients developed thrombocytopenia in the B3 thymoma group, including grades three and four. Nine patients also suffered from thrombocytopenia in the thymic carcinoma group. Anemia manifestation was similar in both groups ($p = 1.000$). None of the patients developed febrile neutropenia. There were no differences in nonhematologic AEs between B3 thymoma and thymic carcinoma groups. Some AEs, such as nausea, vomiting, myalgia, and, arthralgia,

only occurred in a small number of patients. Overall, the rates of AEs were similar in the two groups and both of them were well-tolerated (Table 3).

**Table 3.** Most common Treatment-Related Grade ≥ 3AEs According to NCI-CTCAE.

| AE | B3 Thymoma (*n* = 36) | | Thymic Carcinoma (*n* = 127) | | *p* Value |
|---|---|---|---|---|---|
| | Grade 3 | Grade 4 | Grade 3 | Grade 4 | |
| Hematologic AEs | | | | | |
| Neutropenia | 6 | 5 | 22 | 10 | 0.520 |
| Thrombocytopenia | 3 | 0 | 6 | 3 | 1.000 |
| Anemia | 2 | 0 | 6 | 0 | 1.000 |
| Febrile neutropenia | 0 | 0 | 0 | 0 | 1.000 |
| Nonhematologic AEs | | | | | |
| Fatigue | 3 | 0 | 6 | 0 | 0.672 |
| Sensory neuropathy | 4 | 0 | 8 | 0 | 0.539 |
| Anorexia | 3 | 0 | 5 | 0 | 0.522 |
| Nausea | 2 | 0 | 3 | 1 | 0.861 |
| Myalgia | 1 | 0 | 2 | 0 | 0.530 |
| Arthralgia | 1 | 0 | 2 | 0 | 0.530 |

Abbreviations: AEs, adverse events, NCI-CTCAE, National Cancer Institute Common Terminology Criteria for Adverse Events.

## 4. Discussion

To the best of our knowledge, this is the first study to compare the efficacy and safety of platinum-based chemotherapy between B3 thymoma and thymic carcinoma patients, which evaluated 163 patients with the largest sample research. All patients received the promising efficacy, a difference of efficacy between the two groups did not exist, and received tolerated toxicities. Based on these results, B3 thymoma patients may receive a regimen similar to that used to treat thymic carcinoma patients, especially for thymic squamous carcinoma.

Because of the rarity of thymic epithelial tumors, the related research is limited to a few cases, trials with small sample sizes, and prospective studies. However, platinum-based chemotherapy remains the standard of treatment. The ADOC regimen is the first-line therapy with a reported ORR of 92% and an OS of 15 months [7]. Unfortunately, the heart toxicity of this regimen severely influences its use for thymoma patients [8]. Although the studies on different regimens, such as CAP and EP, have shown that they did not surpass the ADOC for efficacy, further research and exploration have continued. In a study on carboplatin plus paclitaxel use for thymic carcinoma and thymoma patients, the results showed a promising efficacy, demonstrating PFS values for thymoma and thymic carcinoma patients of 16.7 and 5.0 months, respectively [9]. The ORR values were 42.9% for thymoma patients and 21.7% for patients with thymic carcinoma [9]. Furthermore, a phase II study reported that chemo-naïve patients with thymic carcinoma can benefit from carboplatin plus paclitaxel therapy, which exhibited an ORR of 36% and a PFS for thymic carcinoma patients of 7.5 months [10]. The carboplatin plus paclitaxel regimen is regarded as the standard first-line therapy treatment for thymic carcinoma patients. In addition, surgery is the majority therapy method for B3 thymoma, which may influence patient survival. In the present study, the comparison of characteristics between B3 thymoma and thymic carcinoma patients showed a difference that was dependent on whether surgery was carried out (*p* = 0.011). Although the distinction in OS between B3 thymoma and thymic carcinoma was non-existent, the numerical difference in OS may have been attributable to surgery.

The present study results for PFS of 11.3 months in B3 thymoma patients and 10.1 months in thymic carcinoma patients for platinum-based chemotherapy as the first-line treatment confirmed the efficacy of chemotherapy and explored whether the relationship between B3 thymoma and thymic carcinoma may influence the choice of therapy regimen. B3 type

thymoma is defined as a well-differentiated thymic carcinoma (epithelial thymoma or squamoid thymoma), which is characterized by a typically higher stage than other B type thymomas [2]. Some flow cytometry results for CD4 and CD8 single-positive T cells in tumors have revealed that the same molecule was expressed in both B3 thymoma and thymic carcinoma [11]. According to prior research, B3 thymoma is an atypical thymoma with lesions with intermediate features [12]. Compared to thymic carcinoma, B3 thymoma exhibits a less aggressive behavior. Different classifications in early research may have indicated survival with significant prognostic value [13]. According to the complex categories of thymic epithelial tumors, there is an extensive overlap between B3 thymoma and thymic carcinoma [14]. In addition, B3 thymoma is a squamoid thymoma with some similarities to the squamous carcinoma subgroup of thymic carcinoma. Both of them have a common squamous epithelial origin, and the present study showed no differences in their PFS and OS. Therefore, B3 type thymoma and especially squamous carcinoma patients may be treated using a regimen similar to that utilized for thymic carcinoma patients. The other thymic carcinoma subgroup needs further research to analyze its features, correlation with thymoma, and chemotherapy efficacy [15].

The limitations of the present study should also be considered. The small sample sizes of the B3 thymoma group may have caused bias in the analysis. The retrospective nature of study could also influence the analysis results. More clinical experience in the treatment of B3 thymoma patients is also needed. In addition, further prospective study is needed to verify the study outcomes.

## 5. Conclusions

In summary, both B3 thymoma and thymic carcinoma patients can benefit from platinum-based chemotherapy as first-line therapy. The differences in PFS and OS for B3 thymoma and thymic carcinoma patients were not obvious. B3 thymoma patients with a well-differentiated carcinoma and especially squamous carcinoma patients may choose to be treated using a therapy scheme similar to that used for thymic carcinoma patients. This treatment might provide an alternative option for patients in the future and may influence the management of B3 thymoma patients in clinical practice.

**Author Contributions:** Z.S. designed and supervised the research. Y.H., J.S. and J.J. conducted the follow-up, data collection, and correlative analysis. J.W. and J.X. provided support in data analysis and use of software. Y.H., J.S. and C.X. provided data analysis. Y.H. wrote the first draft of the manuscript. All authors have read and agreed to the published version of the manuscript.

**Funding:** The study was supported by the grant from the Foundation of CSCO-Shiyao (Y-SY201901-0068, to Zhengbo Song) and sponsored by Zhejiang provincial program for the Cultivation of High-Level Innovative Health Talents (to Zhengbo Song). China Postdoctoral Science Foundation (2022M723207, to Chunwei Xu) and the Medical Scientific Research Foundation of Zhejiang Province of China (2023KY666, to Chunwei Xu) also provided the support for research.

**Institutional Review Board Statement:** Approval of the study protocol was obtained from Zhejiang Cancer Hospital Institutional Review Board Committee (approval number: IRB-2022-63). Individual consent for this retrospective analysis was waived.

**Informed Consent Statement:** Not applicable.

**Data Availability Statement:** The datasets generated and/or analyzed during the current study are available from the corresponding author on reasonable request.

**Acknowledgments:** The authors would like to thank all patients and their families for their cooperation and participation. Additionally, we are grateful to all research staff and co-investigators involved in this study. Thanks to Hao Huang for enlightening my present and future.

**Conflicts of Interest:** The authors declare no conflict of interest.

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
