# Peer review of "Comparison of Efficacy and Safety of Platinum-Based Chemotherapy as First-Line Therapy between B3 Thymoma and Thymic Carcinoma"

_curroncol, doi:10.3390/curroncol29120743_

Round 1

Reviewer 1 Report

In 5th WHO classification, there is clearly distinction between B3 and thymic cancer. Besides, there are other subtypes than squamous cell carcinoma in thymic cancer.

As the authors mention, since distinction between squamoid thymoma and well differentiated thymic carcinoma is very difficult, it may be significance to investigate clinical implications in both cases.

The niche of this study is based on difficulty of distinguishment between B3 thymoma and squamous cell carcinoma, as such the histologic subclassification of the tumor is mandatory presented that is missed in the manuscript.

Author Response

Dear editor,

Thank you for your kind letter and for the reviewer’s comments concerning our manuscript entitled ‘Comparison of efficacy and safety of platinum-based chemotherapy as first-line therapy between B3 thymoma and thymic carcinoma’. The reviewer’s comments were highly insightful and enabled us to greatly improve the quality of our article. We have studied comments carefully and have made correction which we hope meet with approval. Revisions and point-to-point responses to each of the comments of the reviewers are as follows:

Comments to the author:

Reviewers 1: In 5th WHO classification, there is clearly distinction between B3 and thymic cancer. Besides, there are other subtypes than squamous cell carcinoma in thymic cancer. As the authors mention, since distinction between squamoid thymoma and well differentiated thymic carcinoma is very difficult, it may be significance to investigate clinical implications in both cases.

Reply: Thank you for the high comments in our current study and the affirmation of the value about the clinical implications.

Reviewers 2: The niche of this study is based on difficulty of distinguishment between B3 thymoma and squamous cell carcinoma, as such the histologic subclassification of the tumor is mandatory presented that is missed in the manuscript.

Reply: Thank you for the advice. As we all known, the B3 thymoma had the similar architecture with thymic carcinoma. And the B3 thymoma defined as the squamoid thymoma which implied the correlation with thymic carcinoma, especially for squamous carcinoma. The details about the histologic subclassification of the tumor had been exhibited in the line from 202 to 212.

Reviewer 2 Report

The Authors present the results of a simple, but well-conducted study to compare the effectiveness and safety of platinum-based chemotherapy as first-line therapy between B3 thymoma and thymic carcinoma. The paper is well written and can provide new evidence to enrich the current scientific debate.

I have only some minor concerns:

- some pieces of information is duplicated in the Methods and Results (e.g., date of conduction of the study)

- Figures 1, 2 and 3 look strikingly similar: can the Authors have a look?

- can the Authors expand on the limitations of their study?

- patients were recruited over a very long follow-up period, during which care approaches improved (and a pandemic started). Can this have influenced the results of the study?

Author Response

Dear editor,

Thank you for your kind letter and for the reviewer’s comments concerning our manuscript entitled ‘Comparison of efficacy and safety of platinum-based chemotherapy as first-line therapy between B3 thymoma and thymic carcinoma’. The reviewer’s comments were highly insightful and enabled us to greatly improve the quality of our article. We have studied comments carefully and have made correction which we hope meet with approval. Revisions and point-to-point responses to each of the comments of the reviewers are as follows:

Comments to the author:

Reviewers 1: The Authors present the results of a simple, but well-conducted study to compare the effectiveness and safety of platinum-based chemotherapy as first-line therapy between B3 thymoma and thymic carcinoma. The paper is well written and can provide new evidence to enrich the current scientific debate.

Reply: Thank you for your recognition of this article and the high comments of study.

Reviewers 2: Some pieces of information is duplicated in the Methods and Results (e.g., date of conduction of the study)

Reply: Thank you for your attention. According to the examination of details of study, the concrete part for duplication had been detected. And the duplicated part in the Methods and Results such as date of conduction of the study had been deleted. Thank you for your attention again.

Reviewers 3: Figures 1, 2 and 3 look strikingly similar: can the Authors have a look?

Reply: Thank you for your attention. Our results manifested the similarity of efficacy of platinum-based chemotherapy between B3 thymoma and thymic carcinoma including squamous carcinoma and non- squamous carcinoma. The efficacy of two subtypes of thymic carcinoma had been compared with the B3 thymoma, respectively, which may cause the similarity for figures in appearance. And the target of our study set was to confirm the similarity of efficacy or biological behavior between thymic carcinoma and B3 thymoma which was mentioned above all. So I think your advice was significant. In order to improve the inimitability of our figures, the concrete information of survival for patients had been added.

Reviewers 4: Can the Authors expand on the limitations of their study?

Reply: Thanks for your suggestions. The major limitations of our study are the sample size of research. The limitation of characteristic of retrospective study had been added in the article. Rigorous analytical attitude is worth learning for us. Thank you again.

Reviewers 5: Patients were recruited over a very long follow-up period, during which care approaches improved (and a pandemic started). Can this have influenced the results of the study?

Reply: Thank you for your advice. As we all known, the B3 thymoma had the worse prognosis compared with other types of thymoma including A, AB, B1 and B2 due to the similar biological behavior with thymic carcinoma. And the similarity for B3 thymoma and thymic carcinoma in pathology also implied the similar therapy modes and outcomes. As the advanced thymic epithelial tumors, the care approaches could improve the quality of life, which could not influence the span of life to some extent. So I believe that the long follow-up period could not influence the results of our analysis. Thanks sincerely to you again.

Round 2

Reviewer 1 Report

The revised manuscript considered the subtype of  thymic cancer. Besides, the new version is nicely revised.